# Surgical Treatment of Metastatic Bone Disease in the Appendicular Skeleton: A Population-Based Study

**DOI:** 10.3390/cancers14051258

**Published:** 2022-02-28

**Authors:** Thea Hovgaard Ladegaard, Celine Lykke Sørensen, Rasmus Nielsen, Anders Troelsen, Dhergam Ahmed Ali Al-Mousawi, Rikke Bielefeldt, Michael Mørk Petersen, Michala Skovlund Sørensen

**Affiliations:** 1Musculoskeletal Tumor Section, Department of Orthopedic Surgery, Rigshospitalet, University of Copenhagen, Inge Lehmanns Vej 6, 2100 Copenhagen, Denmark; celine.lykke.soerensen@regionh.dk (C.L.S.); michael.moerk.petersen@regionh.dk (M.M.P.); michala.skovlund.soerensen@regionh.dk (M.S.S.); 2Department of Orthopedic Surgery, Herlev and Gentofte Hospital, Borgmester Ib Juuls Vej 1, 2730 Herlev, Denmark; rasmus.nielsen.06@regionh.dk; 3Department of Orthopedic Surgery, Amager and Hvidovre Hospital, Kettegård Alle 30, 2650 Hvidovre, Denmark; anders.troelsen@regionh.dk; 4Department of Orthopedic Surgery, Nordsjællands Hospital, Dyrehavevej 29, 3400 Hillerød, Denmark; dhergam.ahmed.ali.al-mousawi@regionh.dk; 5Department of Orthopedic Surgery, Frederiksberg and Bispebjerg Hospital, Bispebjerg Bakke 23, 2400 Copenhagen, Denmark; rikke.bielefeldt.01@regionh.dk

**Keywords:** metastatic bone disease, surgery, extremities, appendicular skeleton, biopsy, population-based, cancer

## Abstract

**Simple Summary:**

Patients with bone metastases in the appendicular skeleton (aBM) can experience impending or pathological fractures requiring surgery. Few population-based studies exist and the aim of our retrospective population-based study was to describe a large population of patients surgically treated for aBM, examine changes in incidence of surgery, contrasts between patients at different centers, and the value of tumor biopsies and follow-up imaging. No change in incidence of surgery or absent of sufficient biopsy was found. Significant differences were found between patients treated at different centers. The study enhances the generalizability of our finding to other populations and it is hereby favorable for external validity.

**Abstract:**

*Background*: Population-based studies of patients with bone metastases in the appendicular skeleton (aBM) requiring surgery for complete or impending fracture are rare. In this epidemiologically-based observational study we created a large population-based cohort of patients treated for aBM, aiming to: (1) monitor possible time-related changes of the incidence of surgical treatment of aBM-lesions, (2) examine differences in the population and care of patients treated at different treatment centers and (3) examine if findings from a previous pilot study regarding absence of a suitable biopsy of the lesions representing debut of cancer or a relapse has improved the awareness of aBM and hereby increased the focus on regular tumor biopsies and follow-up imaging of cancer patients. *Methods*: We examined a population-based cohort consisting of all patients treated for aBM 2014–2019. Procedures were performed at five secondary surgical centers (SSC) or one tertiary referral Musculoskeletal Tumor Center (MTC). Patients were followed until end of study (30 September 2021) or death. No patients were lost to follow-up. *Results*: Four-hundred-fifty-seven patients (493 primary aBM-lesions, 482 procedures) were included. Annual incidence of aBM-surgery was 46 aBM-lesions/million. MTC-patients had a significant better preoperative status than SSC-patients considering factors known for survival. Patients with complete fracture experienced longer surgical delay when treated at MTC compared to SSC: 4 (1–9) and 1 (1–3) days (*p* < 0.001), respectively. Overall survival for the entire cohort was 37% and 11% at 1 and 5 years (MTC and SSC 1 and 5 year respectively: 44% and 15% vs. 29% and 5%, *p* < 0.001). In patients with debut or relapse of cancer, 8% and 9% had insufficient biopsies, and 21% and 12% had no biopsy, respectively. Comparison showed no change over time. *Conclusions*: The current study highlights the low awareness on treating aBM at SSC and emphasizes the importance of caution in interpretation of studies not representing an entire population, thus introducing selection bias.

## 1. Introduction

Persons with bone metastases in the appendicular skeleton (aBM) can experience impending or pathological fractures requiring surgery. It is known that bone is the most prevalent site of metastases after the liver and lung [1], however, it is unknown whether surgical intervention for aBM is increasing because of a rise in cancer incidence combined with more patients living longer after a cancer [2].

As this patient group is very heterogeneous, identifying the right implant for the individual patient is essential; especially considering that experiencing an implant failure is devastating. Patients commonly are at their very terminal phase at the time of surgery aiming to preserve quality of life. Therefore, surgeons must consider the patients’ underlying survival prognosis and comorbidities, type of primary tumor, site of metastases and extent of bone loss when determining the most suitable treatment options and which implant will outlive the patient without compromising quality of life [3,4,5].

To date, studies on aBM-surgery are retrospective, include small and selected cohorts from tertiary treatment centers (sampling biased), and often based on historical cohorts or include data over a prolonged period, hence they are confounded by changes in oncological treatment over time. Likewise, most studies are not population-based and consist of heterogeneous populations regarding both demographics and choice of treatment and are therefore heavily biased by selection. Studies on survival of persons with aBM report survival rates ranging 29–70% 1 year after surgery [6,7,8,9,10,11,12,13,14,15,16], indicating that the literature is strongly confounded. This has led to numerous limitations in the examination of what impact orthopedic treatment has on the outcome of aBM patients, including survival, complication rates and implant failures. Many studies comprise cohorts from highly specialized tumor centers and therefore reflect a selected cohort of cancer patients. This minimizes the ability to make generalized conclusions for the entire population regarding the most optimal surgical treatment for patients with aBM. Consequently, larger population-based studies comparing different kinds of aBM-surgery are needed to achieve evidence-based- and improve existing treatment guidelines and enhance the external validity.

Previously, we identified the incidence of surgery for aBM in a 2-year prospective population-based study [17]. We found that it was 48.6 lesions treated per million inhabitants per year, but more interesting we found that the majority of cases representing the debut of cancer or relapse of cancer treated outside a tertiary referral center did not have a suitable biopsy taken of the lesion, which led to a delay in cancer diagnosis and treatment.

In this epidemiologically observational study we sought to create a large population-based cohort of patients treated for aBM, aiming to: (1) monitor possible time-related changes of the incidence of surgical treatment of aBM-lesions, (2) examine differences in the population and care of patients treated at different treatment centers and (3) examine if findings from a previous pilot study [17] regarding the absence of a suitable biopsy of the lesions representing debut of cancer or a relapse has improved the awareness of aBM and hereby increased the focus on regular tumor biopsies and follow-up imaging of cancer patients.

## 2. Materials and Methods

### 2.1. Study Design

We identified a population-based cohort consisting of all patients treated for aBM in the Capital Region of Denmark (CRD) from 1 January 2014–31 December 2019. Patients with hematological disease (myelomatosis and lymphoma) of the bone were included in the study since the same surgical strategy was used. Patients were identified using the regional surgical planning software Orbit or EPIC by looking at all orthopedic surgical procedures in bone in the extremities. For all relevant procedures (approximately 100,000 procedures) we assessed the patient files and evaluated preoperative data, trauma mechanism, preoperative imaging of the fracture/lesion, former cancer history and postoperative follow-up. The appendicular skeleton was defined as the upper and lower extremities including the shoulder girdle and pelvis but excluding hands and feet.

All orthopedic procedures in the CRD, including aBM surgery, was performed at six orthopedic departments: five secondary surgical centers (SSC) (Amager/Hvidovre Hospital, Herlev/Gentofte Hospital, Bispebjerg/Frederiksberg Hospital, Nordsjællands hospitals and Bornholms Hospital) or the tertiary referral Musculoskeletal Tumor Center (MTC) (Rigshospitalet). Due to the highly specialized function at MTC and national treatment guidelines regarding surgery with major bone loss, the majority of aBM-surgeries were refereed to and performed at MTC. The Danish health care system ensured that all patients in the CRD in need of surgery for aBM received treatment at one of these centers. The study hereby comprises a true population-based cohort.

If no biopsy was obtained during surgery or the material was not suitable for histopathological analysis, preoperative pictures, trauma mechanism and postoperative follow-up were considered together. Decision upon inclusion was then made by a team consisting of the primary author and a senior consultant musculoskeletal tumor surgeon and, when appropriate, also a musculoskeletal radiologist.

All surgical procedures were included in the study in case of multiple interventions in one patient during the inclusion period. In case of several aBM-procedures in the same anesthesia all procedures were included with the same baseline and surgical data for all procedures. In case of revision surgery, due to previous aBM-surgery but also other kinds of surgery (e.g., arthrosis or non-pathological fractures) the procedures were excluded (*n* = 40).

### 2.2. Variables

All variables were obtained from the patient files. From the medical records: patient sex, age at index surgery, indication for surgery (complete or impending fractures). A metastasis was considered completely fractured in case of visible lines in cortex at both sides of the bone. An impending fracture was considered a painful bony lesion that was at risk of fracture if no treatment was performed (evaluated by a multidisciplinary team), anatomical location of the metastatic lesion (classified according to the AO classification [18]), presence and number of skeletal and visceral metastases (if no body scans were performed 3 months prior to or after surgery, this variable was considered missing), awareness of aBM-lesion on body scan (lesion described: yes/no), adjuvant therapy status, perioperative biopsy status (reamer dust, biopsy or resection), type of implant used (endoprostheses, osteosynthesis, none) and major bone resection (yes/no). Major bone resection was defined as resection through or below the lesser trochanter at the hip, above the femoral condyles at the knee, below the humeral head, and above the humeral condyles at the elbow, as described previously by Sørensen et al. [19]. Karnofsky Performance Score 30 days prior to surgery was estimated retrospectively by the primary investigator of this study (THL) and was chosen over performance status at surgery time as the presence of e.g., a hip fracture will underestimate the true performance status. Furthermore the American Society of Anesthesiologists (ASA) score was obtained from the anesthesiologist file and ASA score was considered missing if not present here.

From the Danish National Pathology Registry (DNPR) [20] we determined the histopathological diagnosis and date of debut of the cancer. If the primary cancer was unknown and no previous cancer was diagnosed, the date of diagnosis was the date of surgery for aBM. If primary diagnosis was myelomatosis or lymphoma, the date of diagnosis was the first date of registered blood- or bone-marrow related disease. We divided cancer types in three groups, according to the aggressiveness of the cancer: slow, moderate and fast growth cancers. The groups where divided as described in Katagiri et al. [21] and further modification by Sørensen et al. [22].

Patients were followed until end of study (30 September 2021) or until death. No patients were lost to follow-up due to the Danish Civil Registration System which ensures accurate information on emigration and/or death

### 2.3. Statistical Analysis

Descriptive statistic for continues variables was reported as mean (range) or median (IQR) and subgroups were analyzed using Students *t*-test or Wilcoxon-Rank-Sum-test for parametric and non-parametric data, respectively. Test for homogeneity of the variance assumptions was made in order to choose between tests. Chi-squared-test was used for categorical variables. Subgroups being: patients treated at MTC and patients treated at SSC. The same patient could be in both groups in case of multiple aBM-surgeries in the study period. Moods-Median test was used to compare medians between groups for non-parametric variables, subgroups being years and treatment center. Logistic regression was used to describe association. Kaplan-Meier analysis was used for survival analysis for estimating the probability of patient survival with log-rank test for differences. Analysis of patient survival was performed only from index surgery in case of multiple surgeries. This was done to avoid association between limbs or joints in individual patients and hereby potentially biasing results. Confidence intervals were reported as 95% confidence intervals (95% CI) and *p*-values < 0.05 were considered statistically significant. The statistical software R (R Foundation, Vienna, Austria) was used for all statistical analysis.

## 3. Results

### 3.1. Incidence of aBM

In the observation period we identified 457 patients who had a total of 493 primary aBM-lesions and underwent a total of 482 surgical procedures. Thirty-four patients were treated for two lesions and one patient was treated for three aBM-lesions in the study period. Of the 34 patients who were treated for two lesions in the study period, 11 patients were treated for the two lesions during the same anesthesia. Further, two patients had surgery for bone metastases in the axial skeleton during the same anesthesia as the aBM-procedure. We calculated the annual incidence of undergoing surgery for aBM in the CRD by using the mean population of the CRD for each year. Population counts in the study period were extracted from Statistics Denmark [23] and used for the following calculation:$$\frac{\mathrm{aBM}\;\mathrm{lesions}\;\mathrm{treated}\;\mathrm{per}\;\mathrm{year}}{\mathrm{mean}\;\mathrm{population}}\times 1\;\mathrm{mio}\;\mathrm{people}$$

All this resulted in a mean incidence for the entire study period of 45.9 aBM-lesions treated per million inhabitants in the CRD per year. No linear decrease or increase in the overall incidence per year was visually observed (Figure 1).

### 3.2. Patient Demographics and Referral Patterns

The most common primary type of cancer was breast (20%, *n*= 100), lung (20%, *n* = 100), prostate (16%, *n* = 80), kidney (12%, *n* = 59) and myeloma (10%, *n* = 51), accounting for 78% (*n* = 390) of cancer types in the cohort. Unknown primary cancer with no former cancer history in DNPR or previous history in DNPR of the same unknown cancer type was seen in 4.5% (*n* = 22). No differences between patients who were treated at MTC and SSC were identified looking at primary cancer groups (*p* = 0.2). Seventy-nine (16%) lesions were located in the upper extremity and 414 (84%) lesions were located in the lower extremity/pelvis. There was a statistically significant difference between anatomical location of lesion for patients treated at MTC and SSC with a smaller number of lesions located in the upper extremities at SSC (SSC: 11%, MTC: 20%, *p* = 0.01). The most frequent location for metastatic lesions was in the proximal femur counting 63% of the lesions (*n* = 312).

Two-hundred-ninety-six lesions (60%) received an endoprostheses/diaphyseal spacer (*n* = 283/*n* = 13), 174 lesions (35%) received an osteosynthesis and 23 lesions (4.7%) had a resection with no reconstruction. There was a statistically significant difference between patients treated at MTC and at SSC, with the majority of patients at MTC treated with endoprostheses and the majority at SSC treated with osteosynthesis (*p* < 0.001).

Patients who received treatment at MTC had a generally better preoperative status than patients who had treatment at SSC when considering prognostic factors known for better survival. Patients treated at MTC compared to SSC had significant lower age (MTC: 67 years (32–96), SSC: 74 years (43–99), *p* < 0.001), a higher Karnofsky score (MTC: 27% < 70, SSC: 43% < 70, *p* < 0.001), fewer bone metastases (MTC: 69% multiple metastases, SSC: 85% multiple metastases, *p* < 0.001), fewer visceral metastases (MTC: 44% visceral metastases, SSC: 59% visceral metastases, *p* < 0.001) and more patients had an impending instead of a complete fracture (MTC: 66% complete fracture, SSC: 93% complete fracture, *p* < 0.001). We found no differences in aggressiveness of cancer, ASA score, number of days from diagnosis to surgery, if the metastases presented as the debut or relapse of cancer or if the patients received irradiation or medical adjuvant therapy prior to surgery between the two groups. Patient and tumor characteristics are listed in Table 1. Primary cancer type is specified in Table 2.

### 3.3. Biopsies

Twenty percent of the lesions (*n* = 99) represented debut of cancer, 13% (*n* = 66) represented relapse of cancer and 67% (*n* = 328) was in patients living with a known cancer. Of the 99 cases representing debut of cancer, sufficient biopsy material for analysis was obtained in 71% of the lesions (*n* = 70). In 8% (*n* = 8) the biopsy material was insufficient for analysis (material obtained from e.g., reamer dust) and in 21% (*n* = 21) no biopsy during surgery were obtained and of those with no biopsy 17 were treated at SSC. Of the 66 lesions representing relapse of a previous cancer, sufficient biopsy material for analysis was obtained in 79% of the lesions (*n* = 52). Insufficient biopsy was obtained in 9 % (*n* = 6) and no biopsy in 12 % (*n* = 8) and of those with no biopsy seven patients were treated at SSC. Of the 328 lesions in patients living with a known cancer, sufficient biopsy material was obtained in 62% of the lesions (*n* = 203), insufficient biopsy material in 8% (*n* = 27) and no biopsy material in 30% (*n* = 98). Of those with no biopsy, 93 lesions were treated at SSC (Figure 2a–c). Logistic regression showed no change over time in the amount of sufficient biopsies and insufficient/missing biopsies for the entire cohort (*p* = 0.786), for MTC (*p* = 0.138) and SSC (*p* = 0.160).

### 3.4. Surveillance Scan

Of the 394 lesions not representing debut of cancer, 80% of the lesions (*n* = 312) had surveillance scans performed for their primary cancer. Of the 312 lesions, 71% (*n* = 222) of the scanned lesions were described on surveillance scans prior to treatment for aBM, whereas 17% (*n* = 52) were not described prior to surgical treatment. Twelve percent (n = 38) of the lesions were not detected as the surveillance scan did not include the anatomical area of the lesion: 32% (*n* = 12) in the diaphyseal femur, 26% (*n* = 10) in the proximal femur, 16% (*n* = 6) in the diaphyseal humerus, 11% (*n* = 4) in the proximal humerus, 5% (*n* = 2) in the distal femur, 3% (*n* = 1) in the pelvis/acetabular, 3% (*n* = 1) in the tibia/fibula, 3% (*n* = 1) in the distal humerus and 3% (*n* = 1) in the proximal radius/ulna. No linear decrease or increase regarding outcome of surveillance scan per year was visually observed (Figure 3).

### 3.5. Time Till Surgery

The median time from complete fracture of the aBM-lesion to surgery was 2 days (1–7 days) (*n* = 378). We observed differences in median time from fracture to surgery (*p* = 0.001) for patients treated at MTC or SSC: 6 days (2–10 days) and 1 day (1–3 days), respectively (Figure 4). The most frequent site of metastatic lesion in our cohort—the proximal femur (63%, *n* = 312)—had a median time from fracture to surgery at 3 days (1–7 days), again with a statistically significant difference (*p* < 0.001) between MTC and SSC: 7 days (4–11 days) 1 day (1–2 days), respectively (Figure 4).

### 3.6. Survival

At the end of the study period 86% of patients (*n* = 392) had died and 14% (*n* = 65) were still alive. Mean follow-up for the entire cohort of 457 patients was 15.6 months (0–90 months). Mean follow-up for patients who had died was 10 months (0–81 months) while mean follow-up for patients still alive was 49 months (21–90 months). Mean follow-up for the 262 patients in the MTC-cohort was 19.6 months (0–90 months) and for the 195 patients in the SSC-cohort 10.3 months (0–83 months). Mean follow-up for the 274 patients receiving an endoprosthesis was 16.9 months (0–90 months) and for the 163 patients receiving and osteosynthesis mean follow-up was 11.5 months (0–83 months). The cumulated probability for overall survival for the entire cohort at 1, 2 and 5 years after surgery for aBM was 37% (95% CI: 33–42), 26% (95% CI: 22–30) and 11% (95% CI: 8–15) respectively. Kaplan Meier analysis showed a statistically significant difference (*p* < 0.0001) between patients treated at MTC and SSC. Survival at 1, 2 and 5 years was 44% (95% CI: 38–50), 33% (95% CI: 28–39) and 15% (95% CI: 10–21) for MTC and 29% (95% CI: 23–36), 15% (95% CI: 11–21) and 5% (95% CI: 2–10) for SSC. A statistically significant difference (*p* = 0.0022) was also seen between patients receiving an endoprosthesis and an osteosynthesis. Survival at 1, 2 and 5 years was 40% (95% CI: 34–46), 28% (95% CI: 23–34) and 13% (95% CI: 9–19) for endoprostheses and 30% (95% CI: 24–38), 18% (95% CI: 13–25) and 4% (95% CI: 1–12) for osteosyntheses (Figure 5a–c).

## 4. Discussion

In this retrospective population-based study we sought to determine time-related changes in the incidence of surgical treatment of aBM. We found an overall incidence of 45.9 aBM treated per million inhabitants in the CRD per year; however, we did not see an increase over time in our study period. It is probably because the length of the study period was insufficient to identify a change. To the best of our knowledge, only one population-based study [17] on surgical treatment for aBM exists and meaningful comparison of timely changes is therefore difficult and limited. Sørensen et al. [17] conducted their study on parts of the same population as in the present study and found a similar incidence of surgical treatment (48.6 aBM-lesions treated per million inhabitants per year).

Cancer is the second leading cause of death worldwide [24] and we know from several studies that cancer incidence is rising due to increasing age, population growth, and improved diagnostics and screening methods [2,25,26]. With an increasing cancer incidence, the frequency of bone metastases has followed accordingly [2] and we therefore expected a rise in the incidence of surgical treatment of bone metastases in the extremities. Alternatively with the increasing incidence of surgical treatment, one could assume that patients live longer with known aBM before they need surgery. This was shown in a previous study by Hovgaard et al. [8]. Improvements in diagnostics and increased use of modern targeted bone stabilizing therapy [27] prolong time from diagnosis to skeletal-related events [28,29]. Even though adjuvant therapy has advanced, several studies did not find improvement in overall survival [30], indicating that patients needing surgical treatment of aBM are often at an advanced disease level and end of life-stage with a very poor prognosis.

Our study showed a probability of an overall survival of 37% one year after treatment for aBM, which is comparable to other studies regarding this field of patients reporting survival rates ranging 29–70% one year after surgery [6,7,8,9,10,11,12,13,14,15,16]. The differences in the overall survival show that the literature is strongly influenced by selection bias, both according to treatment strategy and general health status of patients included in the studies. We also found a marked difference in overall survival 1, 2 and 5 years after surgery between patients treated at a highly specialized center and those treated at a secondary treatment center. Patients who were treated at MTC had a longer survival, indicating selection bias, when the decision was made regarding which patients to refer for treatment at a highly specialized center. Further we saw differences between patient survival according to which implant was chosen for each patient. Patients who received endoprostheses had a significantly better survival, than patients who received an osteosynthesis. This difference could probably be explained by confounding of treatment center and differences in indication for the chosen implant. The few patients who did not receive an implant during surgery were excluded from this analysis, due to the heterogeneity of the group and hereby differences in indication for surgery. The group included girdlestones, pelvic resections and clavicular resection. As a group they were superior in survival to both endoprostheses and osteosyntheses. Likewise, the different survival rates reflect individual treatment policies for indication of surgery at the different institutions. Several studies showed that impending pathological lesions are more favorable for survival compared with pathological fractures [7,12,31]. This was reflected in our cohort, with a greater number of patients treated at MTC who received treatment for impending fractures (34% compared with 8%), introducing lead time bias in survival analysis.

Discovery of lesions before fracture is preferable. Imaging diagnostics and -follow-up, of both patients with previously known cancer and patients living with a known cancer, is recommendable. This is specially the case in types of cancer known to metastasize to bone, such as breast, prostate and lung cancer [2,32,33], which are also the most common types in our cohort and in the population of Denmark in general [34]. One could advocate for more awareness of skeletal-related events and imaging follow-up. In our study, 21% of the lesions were not followed with any follow-up scan for their primary cancer. Further, we found that 12% of the scans in cancer patients not representing debut of cancer were insufficient as they did not include the particular part of the extremity in the scan area; the majority were in the diaphyseal or proximal femur. We suggest that including these parts in regular scans will lead to early detection, eventually reducing emergency intervention for painful fractures and ultimately improving function and quality of life. Further, metastasectomy for solitary/oligo metastases may improve survival for selected patients [35].

Previously, colleagues investigated the necessity of a biopsy of the lesion during surgery [17]. In the present study, we found that 21% of lesions representing debut of cancer were not biopsied during surgery. Further, we found that in 8% of the lesions the material was insufficient for analysis. This was applicable to both relapse of cancer and for patients living with a known cancer of whom 12% and 30%, respectively, did not have a biopsy of the lesions. Modern treatment of cancer is largely moving towards molecular targeted medicine or “personalized medicine” [36]. In addition, approximately 19% of cancers diagnosed today occur among individuals with a history of previous malignancy [37]. Thus, securing sufficient biopsy material during surgery for histopathological examination is essential. With this we could potentially eliminate whoops procedures, avoid delay in diagnosing relapse or second cancer and ensure targeted therapy. We had expected to see a change during our study period toward fewer insufficient and lacking biopsies; however, this was not the case. A need for more awareness of atypical fractures, concern of aBM and securing proper biopsies for histopathological examinations is essential, especially at SSC, since the majority of lesions without any biopsy were found here.

We observed a difference between centers in the time from fracture of the aBM-lesion until the patients received surgery. In general, time from fracture to surgery was shorter at SSC. In Denmark, the common treatment policy for proximal femoral fractures recommends a maximum of 24 h [38] from the patient entering the hospital until receiving surgery since several studies have shown that surgery delay is crucial to patient survival [39,40,41]. We examined the surgery delay for metastatic lesions in the proximal femur in our cohort and found a median time from fracture to surgery of 7 days at MTC and 1 day at SSC, thereby possibly introducing lead time bias, as mortality might increases when waiting for surgery. One could then ask: is a rushed decision eventually the right decision? The difference between the two groups in surgery delay could possibly be explained by the time spend by transferring the patient from SSC to MTC. Further, the surgical solutions selected at the MTC usually require more planning (often major surgery using mega-prostheses) but also by the organization at the centers. However, we do not find the delay is explained only by the time spent transferring the patient from SSC to MTC. At SSC, fractures are included in the standard trauma regime and are highly prioritized in the surgical planning. At MTC, surgery is performed at a highly specialized center with lack of both prioritization and surgical capacity at the operating rooms coupled with a frequent lack of nursing staff. There is a need to improve capacity at MTC to ensure the risk of increased mortality is reduced. Likewise, there is a need to improve quality of life for the patient bed-bound for 7 days awaiting surgery for a pathological hip fracture

We observed that patients treated at MTC have a better health and preoperative status according to different known prognostic survival factors [5,12,21,22,42] such as lower age, higher Karnofsky score, fewer bone and visceral metastases, and presence of a complete pathological fracture. These findings are in line with other studies examining patients treated at highly specialized centers.

Although these patients are in better health status than those treated at SSC, the overall health conditions of this patient group are poor and expected lifetime is often limited. In the process of determining the most suitable treatment options, surgeons must consider all aspects. In general, patients with short life expectancy can benefit from a less invasive procedure or no surgery at all, whereas those with longer survival estimates often benefit from a resection and reconstruction [6,9,12,43,44]. The surgical treatment strategy was reflected in the type of surgery chosen for each patient in our cohort. The general policy at the MTC facility is a multidisciplinary case-by-case decision, considering all the previously mentioned factors. The majority of patients at MTC were treated with endoprostheses (85%) compared with 26% at SSC. Further, 9.1% and 71%, respectively, were treated with internal fixation at the two treatment centers.

There are limitations to our study. We included patients with all types of primary cancers, thereby including cancers which vary in tendency to metastasize to bone and thus making our cohort extremely heterogeneous. Our study is not suitable for conclusions for any one type of primary cancer but only for a group of all patients with aBM. The retrospective design of the study and the decision to include all cancer types treated surgically introduced some significant limitations, since it was not possible to use the most recently suggested prognostic molecular markers and PDL 1. Instead we used the classification of tumors according to Katagiri et al. [21] and modified by Sørensen et al. [22], that split tumors into slow/moderate/fast growth, which is an old and less precise conception. Further, we examined only patients receiving surgical treatment in the appendicular skeleton, thus excluding patients treated with adjuvant therapy alone and patients with metastases in the axial skeleton. This introduced selection bias. Moreover, we did not have information on postoperative radiotherapy of the lesions, but as orthopedic surgeons we most often do not have influence on postoperative radiation. In general in Denmark, the departments of radiation oncology do not find any indication to treat patients with radiation therapy after surgical treatment of aBM, regardless of the surgical margin obtained. Despite the limitations, there are major advantages of this study. To our knowledge, the study comprises the largest population-based cohort of patients to date receiving surgical treatment for aBM. Consequently it contributes to an improved understanding of aspects concerning the treatment of this patient group. We managed to eliminate selection bias, something that many single center studies struggle with when including only patients treated at highly specialized centers; those patients are often in better general health and have a better preoperative status and thereby more likely to receive an endoprostheses and have longer postoperative survival. The issue of discrepancy between data in scientific literature and larger-scaled registers is also seen in other aspects of orthopedics. The work of D’Ambrossi et al. [45] focuses on total ankle replacement and highlights the differences and biases which is introduced in studies using only selected cohorts from secondary industry sponsorships or highly specialized centers. Hereby it may lower the quality of the scientific results, since the studies do not reflect a true population. Our study enhances the generalizability of our findings to other populations with similar demographics as those of the Danish population making it favorable for external validity and will eventually improve the treatment of this patient group.

## 5. Conclusions

Prevalence of surgical treatment for aBM remains significant and is not a rare event, not even in SSC. No changes in overall incidence of aBM-surgery in the CRD were observed. Further, no change was seen regarding absence of a sufficient biopsy, hence the awareness of this critical matter still remains low and woops procedures may still be an important problem for this patient group. Securing of proper biopsies for histopathological examinations is essential, especially at SSC, since the majority of lesions without biopsies were found here. Biopsies are valuable to exclude a second malignancy and as material for targeted oncological treatment postoperatively. A need of more awareness of atypical fractures and regular imaging follow-up is preferred, since no change was seen in the study period. Standardized imaging protocols for cancer patients, including areas of the appendicular skeleton known to be predominant for aBM may lead to early detection, eventually reducing emergency intervention for painful fractures and finally improving quality of life. Significant differences were found between patients treated at different centers and can reflect the great variation in demographics for patients referred for treatment. It emphasizes the importance of caution in interpretation of studies not representing an entire population, hereby introducing selection bias. This study is hereby favorable for external validity.

## Figures and Tables

**Figure 1 cancers-14-01258-f001:**
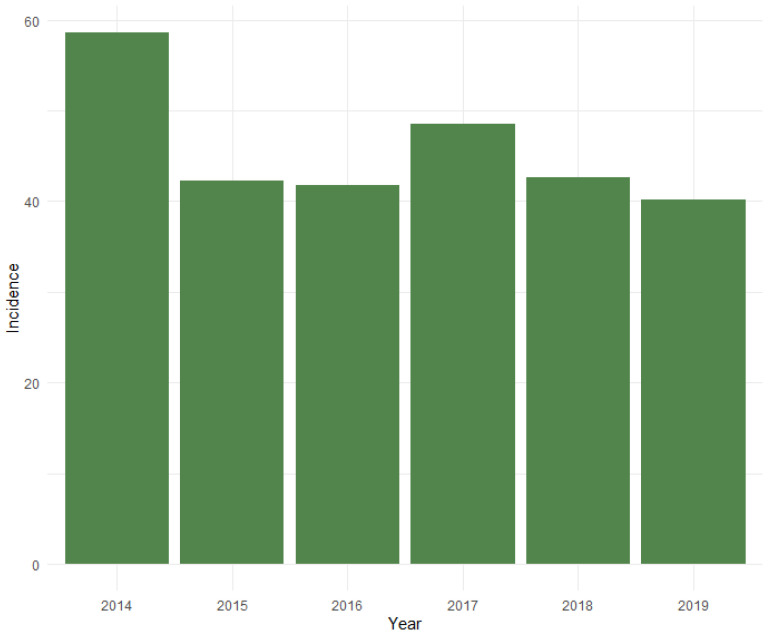
Figure illustrating the incidence of aBM-surgery per million inhabitants in the CRD per year. No linear decrease or increase were seen in the overall incidence of aBM-surgery in the study period.

**Figure 2 cancers-14-01258-f002:**
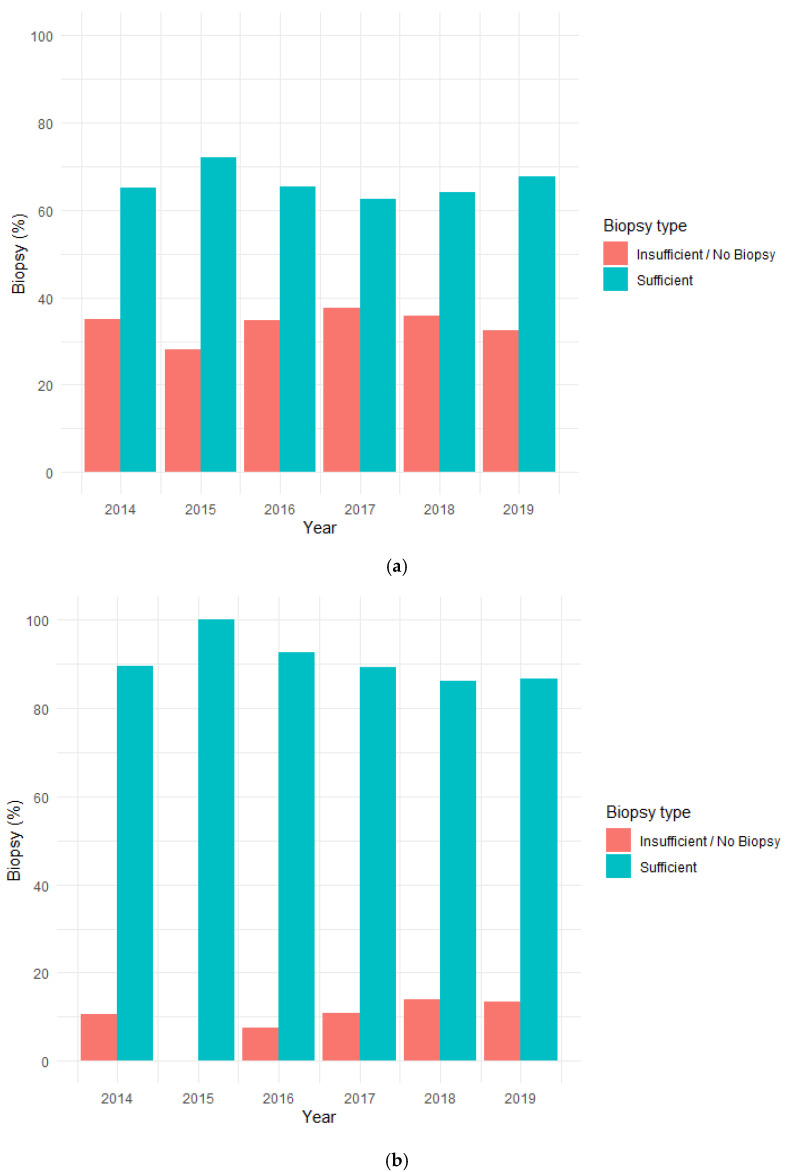
(**a**–**c**). Bar chart illustrating the proportional distribution of biopsies for (**a**) all lesions in the cohort (*n* = 493), (**b**) lesions in the MTC cohort (*n* = 285) and (**c**) lesions in the SSC cohort (*n* = 208). Logistic regression showed no change over time for sufficient biopsies and insufficient/no biopsies for the entire cohort (*p* = 0.786), for MTC (*p* = 0.138) and SSC (*p* = 0.160).

**Figure 3 cancers-14-01258-f003:**
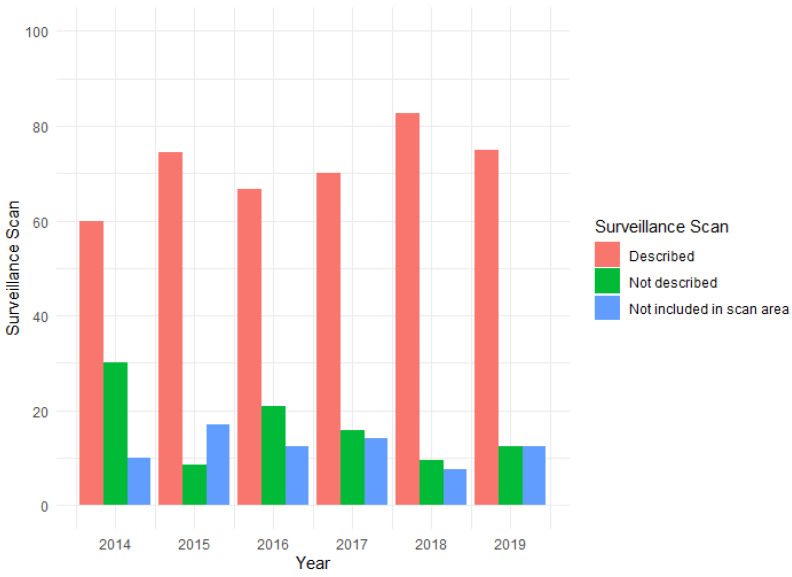
Figure illustrating the distribution of surveillance scans in the entire cohort not representing debut of cancer (*n* = 394). No linear decrease or increase were seen in the distribution of surveillance scans in the study period.

**Figure 4 cancers-14-01258-f004:**
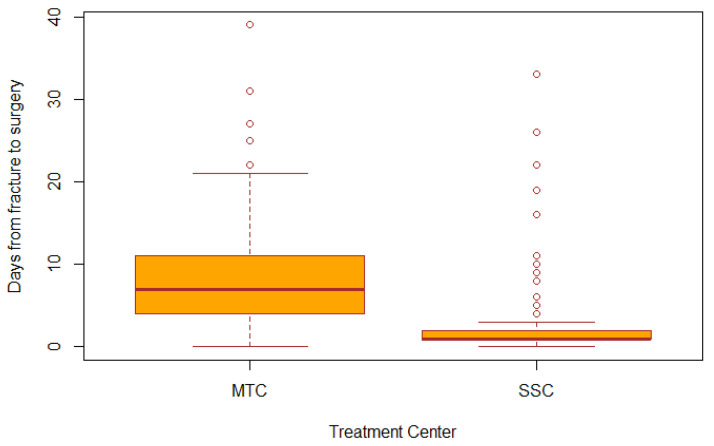
Box plot illustrating median days from fracture to surgery for lesions in the proximal femur. Moods-median-test showed significant difference between MTC and SSC (*p* < 0.001).

**Figure 5 cancers-14-01258-f005:**
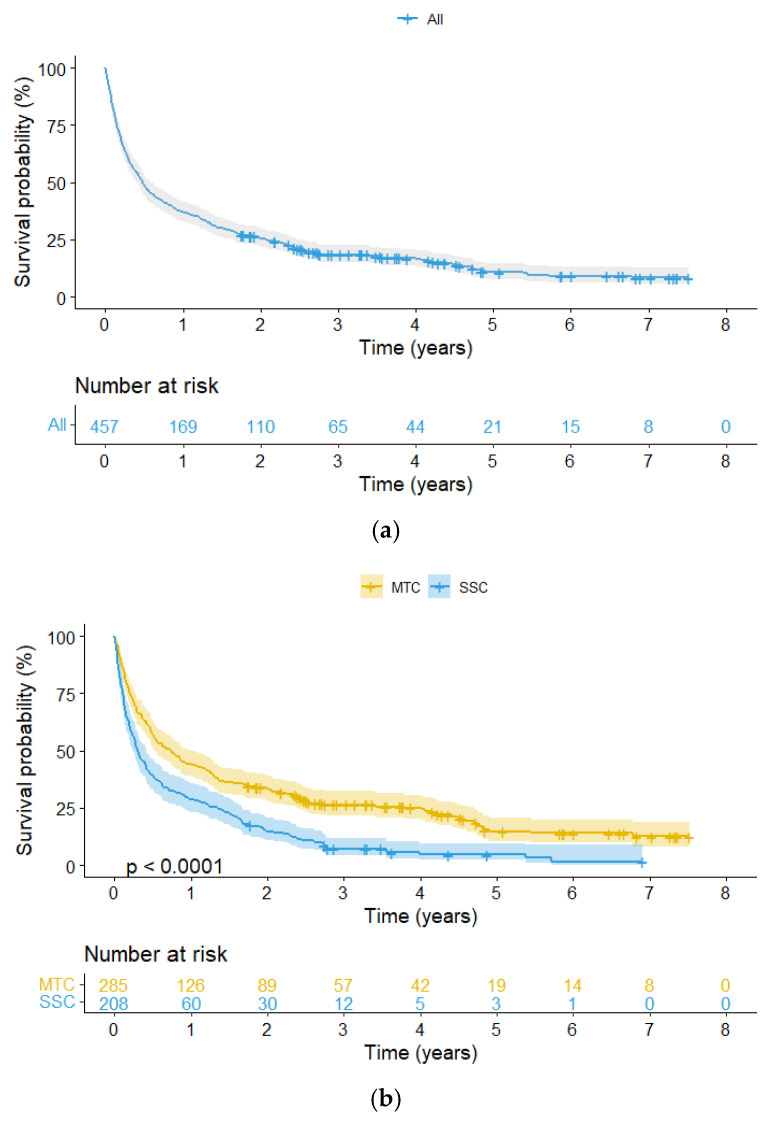
(**a**–**c**) Kaplan-Meier analysis illustrating cumulated overall survival for all patients (*n* = 457). Overall 1-year survival for the entire cohort was 37% (95% CI: 33–42) and 44% (95% CI: 38–50) versus 29% (95% CI: 23–36) for patients treated at MTC or SSC, respectively (*p* < 0.0001), and 40% (95% CI: 34–46) versus 30% (95% CI: 24–38) for patients receiving endoprostheses and osteosyntheses, respectively (*p* < 0.0022).

**Table 1 cancers-14-01258-t001:** Table describing and comparing the preoperative data of the cohort of patients treated for aBM in the CRD per year.

Variable	(*n*/Missing)	All Patients	MTC	SSC	*p*-Value
**Age at Surgery (years)**	(493/0)				<0.001 ^1^
Mean (range)		70 (32–99)	67 (32–96)	74 (43–99)	
**Sex**	(493/0)				0.2 ^2^
Female		254 (52%)	140 (49%)	114 (55%)	
Male		239 (48%)	145 (51%)	94 (45%)	
**Primary Cancer**	(493/0)				0.2 ^2^
Fast Growth		171 (35%)	93 (33%)	78 (38%)	
Moderate Growth		153 (31%)	96 (34%)	57 (27%)	
Slow Growth		169 (34%)	96 (34%)	73 (35%)	
**Location**	(493/0)				0.010 ^2^
Lower Extremity		414 (84%)	229 (80%)	185 (89%)	
Upper Extremity		79 (16%)	56 (20%)	23 (11%)	
**Fracture**	(493/0)				<0.001 ^2^
Complete		378 (77%)	187 (66%)	191 (92%)	
Impending		115 (23%)	98 (34%)	17 (8.2%)	
**Type of Implant**	(493/0)				<0.001 ^2^
Endoprosthesis		296 (60%)	242 (85%)	54 (26%)	
No Implant		23 (4.7%)	17 (6.0%)	6 (2.9%)	
Osteosynthesis		174 (35%)	26 (9.1%)	148 (71%)	
**Karnofsky Score**	(493/0)				<0.001 ^2^
<70		165 (33%)	76 (27%)	89 (43%)	
>= 70		328 (67%)	209 (73%)	119 (57%)	
**ASA Group**	(485/8)				0.074 ^2^
Group 1 + 2		162 (33%)	104 (37%)	58 (29%)	
Group 3 + 4		323 (67%)	180 (63%)	143 (71%)	
**Bone Metastases**	(474/19)				<0.001 ^2^
Solitary Lesion		118 (25%)	89 (31%)	29 (15%)	
Multiple Lesions		356 (75%)	195 (69%)	161 (85%)	
**Visceral Metastases**	(451/42)				0.001 ^2^
No		225 (50%)	152 (56%)	73 (41%)	
Yes		226 (50%)	119 (44%)	107 (59%)	
**Days From Diagnosis to** **Surgery**	(493/0)				0.2 ^1^
Median Days (IQR)		588 (70,2003)	613 (97,2003)	450 (46,2003)	
**Irradiation of Lesion** **Prior to Surgery**	(493/0)				0.063 ^2^
No		411 (83%)	230 (81%)	181 (87%)	
Yes		82 (17%)	55 (19%)	27 (13%)	
**Systemic Treatment**	(493/0)				0.08 ^2^
No		238 (48%)	128 (45%)	110 (53%)	
Yes		255 (52%)	157 (55%)	98 (47%)	
**Debut of Cancer**	(493/0)				0.6 ^2^
No		394 (80%)	230 (81%)	164 (79%)	
Yes		99 (20%)	55 (19%)	44 (21%)	
**Debut of Cancer Relapse**	(394/99)				0.041 ^2^
No		328 (83%)	184 (80%)	144 (88%)	
Yes		66 (17%)	46 (20%)	20 (12%)	

^1^ Wilcoxon rank sum test; ^2^ Pearsons Chi-squared test; MTC: Musculoskeletal Tumor Center; SSC: Secondary Surgical Center.

**Table 2 cancers-14-01258-t002:** Table describing subtypes of cancer divided in subgroups as described in Katagiri et al. [21] and further modification by Sørensen et al. [22].

			Treatment Center	
Primary Cancer	(*n*)	All Patients	MTC	SSC
**Slow growth**	(169)			
Breast		100 (20%)	56 (20%)	44 (21%)
Myeloma		51 (10%)	31 (11%)	20 (9.6%)
Lymphoma		13 (2.6%)	5 (1.8%)	8 (3.8%)
Thyroid		5 (1.0%)	4 (1.4%)	1 (0.5%)
**Moderate growth**	(150)			
Prostate		80 (16%)	48 (17%)	32 (15%)
Kidney		59 (12%)	40 (14%)	19 (9.1%)
Sarcoma		7 (1.4%)	5 (1.8%)	2 (1.0%)
Other gynecological cancer		4 (0.8%)	2 (0.7%)	2 (1.0%)
**Fast Growth**	(174)			
Lung		100 (20%)	52 (18%)	48 (23%)
Colorectal		12 (2.4%)	7 (2.5%)	5 (2.4%)
Malignant Melanoma		9 (1.8%)	7 (2.5%)	2 (1.0%)
Bladder		9 (1.8%)	6 (2.1%)	3 (1.4%)
Head and neck		6 (1.2%)	4 (1.4%)	2 (1.0%)
Pancreatic		5 (1.0%)	2 (0.7%)	3 (1.4%)
Hepatocellular		3 (0.6%)	2 (0.7%)	1 (0.5%)
Gastrointestinal		2 (0.4%)	1 (0.4%)	1 (0.5%)
Gallbladder		2 (0.4%)	1 (0.4%)	0 (0%)
Ventricular		1 (0.2%)	2 (0.7%)	0 (0%)
Unknown origin		22 (4.5%)	9 (3.2%)	13 (6.2%)
Others		3 (0.6%)	1 (0.4%)	2 (1.0%)

## Data Availability

Data is not publicly available, but we will gladly send it on request.

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
