# Peer review of "Surgical Treatment of Metastatic Bone Disease in the Appendicular Skeleton: A Population-Based Study"

_cancers, 2022, doi:10.3390/cancers14051258_

Round 1
Reviewer 1 Report
The article has been significantly improved in the new presentation by adding Table 2, a clearer introduction, more powerful discussion, and more specific conclusions. Thus, the results and conclusions are pertinent, and in my opinion the article can be published
Author Response
Please see the attached Cover Letter.

Reviewer 2 Report
The manuscript has been clearly improved from the intial version. English/typos should be checked again.
In the limitations: add that information about radiotherapy after surgery (on the prosthesis) are not available.
Also add that Sorensen split tumors into slow/moderate/fast growth and that it is an old and insufficient conception with the rise of molecular marker and PDL1 that have considerably changed prognosis in the subgroups of each tumor. Acknowledge that the statistics were limited and that no multivariate survival Cox model has been performed. This is a deep limitation. What is the reason for not performing adjusted multivariate Cox survival model?
Otherwise I am fine with the proposed modification and ok for publication
Author Response
Thank you very much for reading the article once again and taking it into consideration for publishing in your journal.
We have read all the comments from the reviewers and highly appreciate the time and effort they have put into it.
We have followed the advice of Reviewer 2 and added information on postoperative radiotherapy and also about subgrouping of the cancer types under “Limitations”. All changes are made with “track and trace” and marked in the manuscript file.
Regarding the Cox regression analysis we assume that it concerns the change in incidens of aBM during the study period and also changes in biopsies and surveillance scans. As already described in the paper no linear decrease or increase in the overall incidence of aBM per year was visually observed, therefor no statistical analysis was performed. As for evaluation regarding biopsy and surveillance scans, we chose a linear regression method opposed to cox regression. This approach was based on statistician advice (from Brice Maxime Hugues Ozenne, assistant professor at Section for Biostatistics, University of Copenhagen, and several other statisticians from that department), partly based on assumption of proportionality is not expected to be met on our data.
With this in mind, we hope you will find our paper acceptable for publishing in your journal.
Kind regards
Thea H. Ladegaard

This manuscript is a resubmission of an earlier submission. The following is a list of the peer review reports and author responses from that submission.
Round 1
Reviewer 1 Report
The purpose of the work, as well as the results and conclusions are very poor.
The authors propose, as stated in the introduction lines 50-52 "it is unknown whether surgical intervention for MBD is increasing because of a rise in cancer incidence combined with more patients living longer after a cancer"
No conclusion of interest is reached. The only conclusion is that this work “emphasizes the importance of caution in interpretation of studies not representing an entire population, hereby introducing selection bias. This study is hereby favorable for external validity". On the other hand, a statistical study is of little use in this matter, since each patient is a different case.
This work does not serve the stated purpose either. Knowing whether the lower mortality (if any) is due to surgery or not is a matter of no particular importance. In general, surgery - when possible - can greatly alleviate the condition of a cancer patient and could even help slow or prevent metastases by rebuilding the bone.
Author Response
The purpose of the work, as well as the results and conclusions are very poor. The authors propose, as stated in the introduction lines 50-52 "it is unknown whether surgical intervention for MBD is increasing because of a rise in cancer incidence combined with more patients living longer after a cancer" No conclusion of interest is reached. The only conclusion is that this work “emphasizes the importance of caution in interpretation of studies not representing an entire population, hereby introducing selection bias. This study is hereby favorable for external validity". On the other hand, a statistical study is of little use in this matter, since each patient is a different case. This work does not serve the stated purpose either. Knowing whether the lower mortality (if any) is due to surgery or not is a matter of no particular importance. In general, surgery - when possible - can greatly alleviate the condition of a cancer patient and could even help slow or prevent metastases by rebuilding the bone.
The aim and conclusion have been made clearer. Our intention with the article is now clarified in 3 main purposes and all purposes are followed up and answered in the conclusion in the end.
Reviewer 2 Report
Ladegaard et al. present here an interesting retrospective manuscript about the surgery of appendicular bone metastases. It is a multicentric study in a single Danish region. Because of the national health registration they reach a fine exhaustivity of the data. The key messages are: the stability of incidence appendicular bone mets in this region, the difference of population and care between the reference musculoskeletal center and the peripheral ones, the lack of biopsy in the peripheral centers.
Comments:
- They use de MBDex for Metastatic Bone Disease Extremities but they do not define the concerned regions sufficiently early in the manuscript. Does it refer only to hands, fingers, ankles and feet? If yes, this term is suitable otherwise (Humerus, femur, tibia, radius), it would be more appropriate to name them appendicular bone mets. Please clarify, define the perimeter of the study in the introduction and abstract and adapt the abbreviation throughout the manuscript (aBM).
- What is the percentage of radiation therapy performed on the fracture bone after orthopaedic surgery in the whole population and according to SSC/MTC?
- Please add details on cancer histology in table 1 (and eventually molecular status) in baseline description: melanoma, kidney, breast, prostate, lung (epidermoid, adenocarcinoma), thyroid etc…
- Are there any surgery differences (techniques etc…) between died vs alive patients?
- How was define the fast/moderate/slow growing tumor group? Please clarify these criteria in the methods.
- Table 1: lines on ASA group can be simplify since they do not bring anything in particular. I suggest to delete lines ASA 1, 2, 3,4 and keep only the 2 last lines.
- The authors should take some conclusions based on their study regarding: initial scan detection and scan surveillance recommendation.
Author Response
Ladegaard et al. present here an interesting retrospective manuscript about the surgery of appendicular bone metastases. It is a multicentric study in a single Danish region. Because of the national health registration they reach a fine exhaustivity of the data. The key messages are: the stability of incidence appendicular bone mets in this region, the difference of population and care between the reference musculoskeletal center and the peripheral ones, the lack of biopsy in the peripheral centers.
Comments:
They use de MBDex for Metastatic Bone Disease Extremities but they do not define the concerned regions sufficiently early in the manuscript. Does it refer only to hands, fingers, ankles and feet? If yes, this term is suitable otherwise (Humerus, femur, tibia, radius), it would be more appropriate to name them appendicular bone mets. Please clarify, define the perimeter of the study in the introduction and abstract and adapt the abbreviation throughout the manuscript (aBM).
We now define the appendicular skeleton in section 2.1 “Study Design” and further MBDex has been changed to aBM throughout the text.
What is the percentage of radiation therapy performed on the fracture bone after orthopaedic surgery in the whole population and according to SSC/MTC?
This was not one of our primary focuses with the article. As orthopedic surgeons we most often do not have influence on postoperative radiation therefor this was not included in the article. In general, in Denmark the Departments of radiation oncology do not find any indication to treat patients with radiation therapy after surgical treatment of metastatic bone disease of extremities regardless of the surgical margin obtained.
Please add details on cancer histology in table 1 (and eventually molecular status) in baseline description: melanoma, kidney, breast, prostate, lung (epidermoid, adenocarcinoma), thyroid etc…
A more detailed overview of the cancers is now specified in Table 2, although not on a molecular level due to lack of information. Further, within orthopedic oncology we have precedent of presenting it as done in the article and specification on cancer subtypes will not change their classification in the clinically used prediction models.
Are there any surgery differences (techniques etc…) between died vs alive patients?
We have added information on surgical differences in section 3.5 “Survival”. Further, we have commented on the differences between implants and survival status in the discussion.
How was define the fast/moderate/slow growing tumor group? Please clarify these criteria in the methods.
This is now further specified in the section “2.2 Variables”. “We divided cancer types in 3 groups according to the aggressiveness of the cancer: slow, moderate and fast growth cancers. The groups where divided as described in Katagiri et al.[21] and further modification by Sørensen et al.[22].” Additional, all cancer subtypes in the three groups have been defined in Table 2. We hope this answer your question.
Table 1: lines on ASA group can be simplify since they do not bring anything in particular. I suggest to delete lines ASA 1, 2, 3,4 and keep only the 2 last lines.
We have followed the advice and removed ASA 1-4.
The authors should take some conclusions based on their study regarding: initial scan detection and scan surveillance recommendation.
We now include surveillance scan in the conclusion as well as in the discussion section.
Reviewer 3 Report
Dear Authors, I read your paper, appreciating its completeness and usefulness.
I consider it well written and conducted, methodologically and conceptually. In the discussion paragraph, I suggest to deal the concept concerning the importance of dedicated registers for each pathology, mentioning “D'Ambrosi R, Banfi G, Usuelli FG. Total ankle arthroplasty and national registers: What is the impact on scientific production? Foot Ankle Surg. 2019 Aug;25(4):418-424. doi: 10.1016/j.fas.2018.02.016. Epub 2018 Mar 6. PMID: 30321963.”.
Apart from this aspect, in my opinion the paper deserves publication.
Author Response
Dear Authors, I read your paper, appreciating its completeness and usefulness.
I consider it well written and conducted, methodologically and conceptually. In the discussion paragraph, I suggest to deal the concept concerning the importance of dedicated registers for each pathology, mentioning “D'Ambrosi R, Banfi G, Usuelli FG. Total ankle arthroplasty and national registers: What is the impact on scientific production? Foot Ankle Surg. 2019 Aug;25(4):418-424. doi: 10.1016/j.fas.2018.02.016. Epub 2018 Mar 6. PMID: 30321963.”.
Apart from this aspect, in my opinion the paper deserves publication.
We included the mentioned reference in the end of the discussion.